# Increased Temporal Overlap in Diel Activity Patterns Potentially Intensifies Interspecific Competition Among Sympatric Large Carnivores in the Sanjiangyuan Region of China

**DOI:** 10.3390/ani15142059

**Published:** 2025-07-12

**Authors:** Dong Wang, Quanbang Li, Jingyu Gao, Xu Su, Xinming Lian

**Affiliations:** 1School of Geographical Science, Qinghai Normal University, Xining 810016, China; wangdong@nwipb.cas.cn; 2Key Laboratory of Adaptation and Evolution of Plateau Biota, Northwest Institute of Plateau Biology, Chinese Academy of Sciences, Xining 810008, China; liquanbang@nwipb.cas.cn (Q.L.); gaojingyu24@mails.ucas.ac.cn (J.G.); 3University of Chinese Academy of Sciences, Beijing 100049, China; 4Key Laboratory of Biodiversity Formation Mechanism and Comprehensive Utilization of the Qinghai-Xizang Plateau in Qinghai Province, Qinghai Normal University, Xining 810008, China; 5Academy of Plateau Science and Sustainability, Qinghai Normal University, Xining 810016, China; 6Qinghai Provincial Key Laboratory of Animal Ecological Genomics, Xining 810008, China

**Keywords:** carnivores, infrared camera trap, activity time overlap, interspecific interference, sympatric coexistence

## Abstract

Direct competition for resources is particularly pronounced among apex predators, significantly influencing their fitness and functional roles within ecosystems. Complex predator guilds further intensify these competitive pressures through dominance hierarchies that clearly define winners and losers. In the Sanjiangyuan Region (SR) of Qinghai Province, China, a diverse array of apex predators is distributed. Their cryptic and elusive behaviors have resulted in limited understanding of their activity patterns and the extent of interspecific overlap. To investigate the intensity of interspecific competition among large carnivores in the SR from the perspective of temporal niche differentiation, we leveraged approximately a decade of data collected from infrared camera traps. This analysis focused on quantifying the degree of overlap in diurnal activity rhythms across all pairwise combinations of four large carnivore species. Our findings suggest that pairwise diel activity patterns exhibit moderate to high levels of overlap, potentially intensifying interspecific competition among these carnivore species within the region.

## 1. Introduction

Large carnivores (body mass ≥ 15 kg), commonly referred to as apex predators, play a pivotal role in ecosystem conservation by serving as flagship species that enhance public awareness and support fundraising initiatives [1,2]. The effects of apex predators often cascade through food chains and webs, influencing other faunal groups such as birds and invertebrates, and ultimately affecting plant abundance and distribution [3]. This influence permeates throughout the entire ecological community, contributing significantly to the maintenance and formation of biodiversity [4]. Consequently, understanding the behavioral ecology of large carnivores is fundamental to managing public perceptions and attitudes, and promoting human–large carnivore coexistence in multifunctional landscapes [5]. However, due to their wide-ranging distributions and cryptic behaviors, conducting reliable behavioral observations of large carnivores remains a significant challenge [6,7].

Infrared camera technology, recognized for its accuracy, long-term monitoring capability, concealment, and non-invasive nature, is widely employed to investigate activity patterns and interspecific interactions among elusive, low-density large carnivores [8,9]. It records the precise timing, frequency, and duration of animal presence, providing long-term data for analysis. Activity time refers to the period during which an animal is detected by the camera, whereas activity level denotes the proportion of time spent in active behavior [10]. These methodologies are particularly valuable for studying elusive species in remote areas [11].

Variations in daily activity patterns result in the classification of wildlife into four primary categories: crepuscular, diurnal, nocturnal, and cathemeral [12]. Research has demonstrated that animals continuously adjust their diurnal activity patterns and activity peaks throughout evolutionary processes to balance the costs associated with interspecific competition, resource acquisition, and predation risk [11,13]. When competing species co-occur within a limited spatial area, the subordinate species typically modifies its activity rhythm to minimize overlap with the dominant competitor or to avoid habitats that are most favorable for the survival of the dominant species [14,15,16,17]. In cases where sympatric large carnivores exhibit high similarity in spatial, trophic, and functional traits, they often adapt to cyclical environmental changes and competitive pressures by distributing their activity intensity and patterns across different times of day and seasons [18,19]. Therefore, variations in activity rhythms play a crucial role in facilitating coexistence among sympatric large carnivore species.

The Sanjiangyuan Region (SR), characterized by its representative ecosystems and high biodiversity on the Qinghai–Xizang Plateau [20], serves as a critical habitat and genetic repository for endemic wildlife [21]. It maintains high levels of wildlife authenticity and habitat integrity, supporting a diverse array of endemic and rare species, including large carnivores such as snow leopards (*Panthera uncia* Schreber, 1775), wolves (*Canis lupus* Linnaeus, 1758), brown bears (*Ursus arctos* Linnaeus, 1758), and Eurasian lynx (*Lynx lynx* Linnaeus, 1758) [22,23]. Previous studies on these sympatric carnivores have focused on diet, predator–prey dynamics, human conflict, and population density [24,25,26,27]. However, research on competitive intensity and coexistence mechanisms through temporal niche differentiation remains limited [28]. To better understand interspecific competition among sympatric large carnivores in the SR and to inform evidence-based conservation policy, this study utilizes a decade-long dataset composed of data from infrared camera surveys (2014–2024). It analyzes daily activity overlap among snow leopards, wolves, brown bears, and Eurasian lynx, and examines competition intensity and potential coexistence mechanisms via temporal niche differentiation.

## 2. Materials and Methods

### 2.1. Study Area

The SR, located in the heart of the Qinghai–Xizang Plateau and spanning southern Qinghai Province, covers an area of approximately 396,000 km^2^ (31°39′~36°16′ N, 89°24′~102°23′ E; Figure 1). It serves as the source of the three major rivers in China: the Yangtze River, the Yellow River, and the Lancang River [29]. Additionally, it is recognized as a critical biodiversity hotspot and a key recharge zone for freshwater resources in China, earning the nickname “Water Tower of China”. Due to its unique geographical position and significant ecological roles, the SR acts as a vital ecological refuge not only for China but also for the broader Asian continent. The region exhibits complex topography, with an average elevation exceeding 4000 m. Its climate is characterized by plateau continental features, marked by small annual temperature ranges and large diurnal temperature variations. The mean annual temperature ranges from −5.6 °C to 3.8 °C, while annual precipitation varies between 262 mm and 772 mm, with 85% occurring during the months of June to September [30]. Unlike regions with the traditional four distinct seasons, this area is predominantly divided into cold and warm seasons. The main mountain ranges include the Tanggula, Bayan Har, East Kunlun, and Anyemaqen Mountains [31]. Grassland constitutes 74% of the SR, with meadows and steppes being the dominant vegetation types, accounting for 51% and 22% of the region, respectively [32]. The SR boasts rich faunal biodiversity, encompassing 85 species of wild mammals across 20 families and eight orders, 238 bird species across 41 families and 16 orders, and 15 species of amphibians and reptiles [22]. Most of these species are classified as National Key Protected Wildlife in China.

### 2.2. Camera Trapping and Data Pre-Processing

Among the six large carnivore species documented in the SR [22], the snow leopard, wolf, brown bear, and Eurasian lynx are relatively abundant and maintain sympatric distributions within the SR [27]. Between June 2014 and April 2024, six areas within the SR, characterized by frequent activity of snow leopards, wolves, brown bears, and Eurasian lynx, were selected for the deployment of infrared camera traps. Over the entire survey period, a total of 422 infrared cameras (Ltl-L6210 MC, Shenzhen Ltl Acorn Electronics Co., Ltd. Shengzhen, China; ERE-E1B, Shenzhen Ereagle Technology Co., Ltd. Shengzhen, China) were deployed across these areas (Figure 1), covering approximately 2580 km^2^, with monitoring altitudes ranging from 3924 m to 5078 m.

During the placement of camera traps, we first consulted ecological rangers and local herders with extensive knowledge of regional wildlife distributions to identify key areas frequently utilized by and preferred as habitats for the four large carnivore species. Camera deployment locations were determined based on the comprehensive consideration of terrain features and site accessibility. Cameras were primarily installed at strategic locations such as animal trails, saddles, ridge lines, river valleys, water sources, and the bases of large boulders, where encounters with the target species were most likely. Each infrared camera was securely anchored using ground stakes or rocks, maintaining a vertical height of 50–80 cm above the ground [9], with lenses orientated horizontally to minimize direct sunlight exposure.

Additionally, detailed information, including the unique identification number of each camera, its corresponding latitude and longitude coordinates, elevation, and surrounding habitat characteristics, was meticulously recorded. To mitigate potential spatial autocorrelation issues arising from excessive proximity between adjacent camera sites, we ensured that the straight-line distance between neighboring deployment sites exceeded 500 m during field setup [33]. Following installation, camera data were retrieved and maintenance was performed at intervals of approximately three to four months.

After retrieving the infrared camera data, folders were systematically created for each camera site. The Bio-photo V 2.1 software was subsequently employed to automatically extract key information, including the identification number, capture date, and time, for each photograph and video. This information was then exported into an Excel spreadsheet for further processing. Thereafter, the captured photographs and videos were manually reviewed for individual identification, and the images were simultaneously filtered to isolate records of snow leopards, wolves, brown bears, and Eurasian lynx. The criteria for defining independent events were adapted from previous studies. Specifically, photographic captures of the same species at the same camera trap location within a 30 min interval were grouped as a single independent and valid record for subsequent data analysis [9,34]. It should be noted that all infrared camera photographs used in the analysis of diurnal activity rhythms for the four large carnivore species were derived from independent events.

### 2.3. Daily Activity Pattern Analysis

Kernel density estimation (KDE) is a non-parametric statistical method used to estimate the probability density function of discrete observational data [35]. In studies of daily activity patterns, KDE is frequently employed to analyze the temporal distribution of animal activity throughout the day. Through the plotting of density curves, it provides an intuitive visualization of periods of peak and minimal activity, contributing to its widespread application in characterizing animal activity rhythms [8,9,10]. This study employed KDE to quantify the activity intensity of snow leopards, wolves, brown bears, and lynx at each time point, thereby elucidating the diel activity patterns of these four large carnivores [35,36]. The calculation formula is as follows:$$f\left(x;v\right)=\frac{1}{n}\sum_{n=1}^{n}K_{v}\left[d\left(x,x_{i}\right)\right]$$

In this formula, *K_v_* denotes the probability density function of the von Mises distribution, *d* (*x*, *x_i_*) represents the angular distance between an arbitrary point *x* and sample *i*, and the cumulative sum of activity intensities across all time points is normalized to 1. The kernel density estimation method assumes that the probability of an animal being captured by an infrared camera during a specific time interval is proportional to its activity intensity [36].

Based on the defined climatic conditions in the SR, the study period was divided into two distinct seasonal phases: the warm season (May to September) and the cold season (October to April of the following year) [37]. Additionally, considering the substantial variation in sunrise (ranging from 5:30 to 8:30, Beijing time) and sunset (ranging from 17:30 to 21:00, Beijing time) times throughout the year within the study area, all infrared camera capture times were standardized to solar time. This adjustment aimed to mitigate discrepancies in sunrise and sunset times across different seasons [38,39]. The correlation analysis was performed using the “sunTime” function within the “overlap” statistical package in R software (R version 4.4.2).

### 2.4. Activity Time Overlap Analysis

The temporal overlaps between large carnivores were quantified using the overlap coefficient Δ, which is defined as the area under the two kernel density curves and ranges from 0 (no overlap) to 1 (complete overlap) [35,40]. The 95% confidence interval for the overlap coefficients was estimated through bootstrapping with 1000 samples derived from the estimated probability density functions [41]. Additionally, we classified the degree of diel activity rhythm overlap among pairs of large carnivore species during the entire year, cold season, and warm season into three categories based on the magnitude of the overlap coefficient Δ. Specifically, 0 < Δ < 0.5 indicated low overlap, 0.5 ≤ Δ < 0.8 denoted moderate overlap, and 0.8 ≤ Δ < 1 signified high overlap [42]. These three overlap levels corresponded sequentially to low, moderate, and high levels of competitive intensity between the paired large carnivore species.

## 3. Results

From June 2014 to April 2024 (Table 1), a total of 102,913 effective camera days were accumulated across 422 camera sites in six distinct survey regions within the SR through the deployment of varying numbers of infrared cameras and monitoring for different durations (Table 1). Statistical analysis of the infrared camera data revealed that, during the survey period, there were a total of 15,424 valid photographic captures for snow leopards, 1778 for wolves, 3594 for brown bears, and 344 for Eurasian lynx. Specifically, a total of 3312 independent and valid records were obtained for snow leopards, 352 for wolves, 240 for brown bears, and 79 for Eurasian lynx.

### 3.1. The Degree of Daily Activity Rhythm Overlap Throughout the Year

The analysis of the annual diel activity rhythm curves indicated that all four large carnivores predominantly exhibit nocturnal behavior, with notable differences observed in their peak activity periods (Figure 2, Table 2). A statistical evaluation of the annual diel activity rhythm overlap coefficient between pairwise species revealed that the highest overlap coefficient occurred between snow leopards and Eurasian lynx (Δ = 0.88, 95% confidence interval (CI): 0.80–0.95, Figure 2c), followed by snow leopards and wolves (Δ = 0.86, 95% CI: 0.82–0.90, Figure 2a), as well as wolves and Eurasian lynx (Δ = 0.83, 95% CI: 0.74–0.92, Figure 2e). These three pairs of species exhibited a high level of overlap in their diel activity rhythms. Additionally, the diurnal activity rhythms between snow leopards and brown bears (Δ = 0.71, 95% CI: 0.65–0.77, Figure 2b), wolves and brown bears (Δ = 0.67, 95% CI: 0.60–0.74, Figure 2d), and brown bears and Eurasian lynx (Δ = 0.66, 95% CI: 0.56–0.76, Figure 2f) demonstrated a moderate level of overlap for each pairwise species combination.

### 3.2. The Degree of Daily Activity Rhythm Overlap During the Cold Season

From the perspective of the cold season (Table 2), the diel activity rhythm curves of the pairwise species pairs between snow leopards and wolves (Δ = 0.86, 95% CI: 0.82–0.91, Figure 3a), snow leopards and Eurasian lynx (Δ = 0.86, 95% CI: 0.78–0.94, Figure 3c), and wolves and Eurasian lynx (Δ = 0.85, 95% CI: 0.75–0.94, Figure 2e) still exhibited high-density overlap. In comparison with the annual pattern, the degree of overlap in the diurnal activity rhythms between snow leopards and brown bears (Δ = 0.70, 95% CI: 0.62–0.77, Figure 3b), wolves and brown bears (Δ = 0.65, 95% CI: 0.55–0.74, Figure 3d), and brown bears and Eurasian lynx (Δ = 0.60, 95% CI: 0.48–0.72, Figure 3f) during the cold season shows a slight decline, yet all still fall within the range of moderate overlap.

### 3.3. The Degree of Daily Activity Rhythm Overlap During the Warm Season

It is noteworthy that all species pairs exhibited moderate overlap in their diurnal activity rhythms during the warm season (Figure 4, Table 2). When compared with the degrees of overlap in diurnal activity rhythms throughout the year and during the cold season, the overlap coefficients between snow leopards and wolves (Δ = 0.79, 95% CI: 0.70–0.87, Figure 4a), snow leopards and Eurasian lynx (Δ = 0.73, 95% CI: 0.55–0.88, Figure 4c), and wolves and Eurasian lynx (Δ = 0.66, 95% CI: 0.47–0.84, Figure 4e) during the warm season all decreased. There was little change in the overlap of diurnal activity rhythm curves between snow leopards and brown bears (Δ = 0.69, 95% CI: 0.61–0.77, Figure 4b), wolves and brown bears (Δ = 0.61, 95% CI: 0.50–0.72, Figure 4d), and brown bears and Eurasian lynx (Δ = 0.69, 95% CI: 0.51–0.85, Figure 4f) during the warm season when compared with the overlaps throughout the year and during the cold season.

## 4. Discussion

Globally, large carnivore populations are limited by their inherently low population densities [43]. As human activities, such as livestock grazing and logging operations, continue to expand spatially, the amount of suitable habitat available for large carnivores has been progressively diminished [44]. High-altitude mountain ecosystems, despite often harboring high levels of biodiversity and endemism, present significant challenges for large carnivores due to harsh environmental conditions, low productivity, scarce resources, and anthropogenic habitat compression. These factors may result in reduced ecological niche space among sympatric species [45,46]. Consequently, investigating how sympatric carnivores partition and utilize limited alpine resources, as well as examining their interspecific interactions and coexistence mechanisms, is critical for understanding the structuring and long-term persistence of biodiversity within these unique ecosystems.

This study utilized an extensive network of camera traps deployed across the SR within the hinterland of the Qinghai–Xizang Plateau over multiple years, systematically collecting activity data on four large carnivore species: snow leopards, wolves, brown bears, and Eurasian lynx. We analyzed the degree of overlap in diurnal activity rhythm among these species to assess the intensity of competition among sympatric large carnivores from the perspective of temporal niche differentiation. These findings might inform practical conservation and management strategies for maintaining the ecological integrity of the large carnivore guild in this region. Despite moderate or high overlap in pairwise comparisons of diel activity rhythms among the four species, our results indicated considerable variation in their peak activity periods. This variation partially alleviated potential interspecific competition in the temporal niche dimension. Based on the existing literature, the activity rhythms of large carnivores were primarily influenced by the activity patterns of their primary prey [47]. Moreover, differences in prey activity patterns represented a key factor shaping the variations in carnivore activity rhythms [48,49]. Therefore, we hypothesized that the distinct diurnal activity peaks observed among the four carnivore species may be linked to differences in the activity patterns of their preferred prey.

Research has indicated that interspecific competition was a primarily driver of resource partitioning across three niche dimensions: spatial, temporal, and trophic [8,50]. A species can enhance its persistence stability within a community by differentiating along at least one of these niche dimensions [18]. Compared to spatial and trophic niche differentiation, temporal niche differentiation exhibits greater flexibility and adaptability [19], effectively reducing competitive suppression among sympatric carnivores with overlapping ecological traits [51]. In this study, snow leopards and Eurasian lynx exhibited the highest degree of overlap in their diurnal activity rhythms (Δ = 0.88). Both species displayed peak activity around sunrise and sunset, indicating substantial potential for interspecific interference in terms of temporal niche utilization. However, while high niche overlap serves as an indicator of potential competition intensity between species within a community, it does not fully equate to the actual strength of interactions. Significant temporal overlap primarily reflects an increased likelihood of encounters between sympatric carnivore species within the same area but does not necessarily imply synchrony in their activity patterns [52,53]. Although snow leopards and Eurasian lynx shared similar peak activity periods, the lynx exhibited a more prolonged activity peak, with its sunrise-related activity distinctly delayed relative to the snow leopard. This temporal segregation likely serves to reduce direct competition between the two species [8]. It also indicates that temporal niche differentiation exhibits greater flexibility and adaptability compared to other niche dimensions, thereby effectively minimizing interference among sympatric carnivores with similar ecological requirements [19].

Based on the existing literature, spatial utilization patterns exhibit significant variation among these species [54,55,56,57,58]. Snow leopards predominantly inhabit high-altitude ecosystems above 3000 m in elevation, demonstrating a strong preference for rugged terrains such as ridgelines, narrow mountain passes, and canyons [54,55]. Wolves, being the most widely distributed large carnivore globally, display remarkable ecological adaptability and are commonly found across a wide range of ecosystems, including forests, shrublands, grasslands, wetlands, rocky areas, and deserts [56]. Brown bears, as opportunistic foragers, typically select habitats such as large rock formations or natural dens on hillsides to optimize hunting efficiency [57]. The Eurasian lynx is well adapted to cold climates at high altitudes and primarily occupies various habitats, including forests, river valleys, grasslands, and deserts [58]. In terms of prey utilization, although all four large carnivores primarily prey on ungulates, they differ significantly in terms of prey species selection and proportional consumption rates [8,24]. Wolves, employing a group-hunting strategy, target the largest prey, whereas brown bears, as opportunistic omnivores, tend to focus on smaller prey items [24]. These differences collectively contribute to trophic niche differentiation among the four species, facilitating their sympatric coexistence.

Research demonstrates that large carnivores often exhibit considerable overlap in activity patterns, with interspecific competition primarily mitigated through spatial niche differentiation, thereby enabling regional coexistence [8]. This is consistent with the findings of the present study, which revealed a high degree of overlap in diel activity rhythms among the four focal large carnivore species. Regional coexistence among carnivores represents a relatively stable ecological state shaped by long-term natural selection during evolutionary processes. Consequently, no single mechanism can fully explain the assembly rules governing diverse carnivore communities. Although this study collected extensive infrared camera data through multi-year monitoring, a comprehensive understanding of the mechanisms underlying large carnivore competition and coexistence cannot be derived solely from analyses of temporal niche differentiation. Future research should integrate investigations into spatial and trophic niche differentiation while considering additional factors such as prey abundance and community composition, habitat quality and distribution, and the types and intensities of anthropogenic disturbances affecting species’ temporal, spatial, and trophic niches. Such an integrative approach will facilitate a more holistic understanding of the mechanisms driving the regional coexistence of large carnivores in high-altitude ecosystems.

## 5. Conclusions

Currently, approximately 71% of large carnivore populations globally are experiencing continuous declines, and 60% of these species now occupy less than half of their historical ranges, thereby increasing the risks of local or global extinction [1]. When a large carnivore population declines or is locally extirpated, it inevitably results in the simplification of food web structure and compromises the stability of ecological communities [59]. However, a limited understanding of the activity patterns of large carnivores has hindered comprehensive research into their interspecific relationships. Through multi-year monitoring, this study demonstrated that snow leopards, wolves, brown bears, and lynx predominantly exhibited nocturnal activity, although significant differences existed in their peak activity periods. Furthermore, while pairwise diel activity rhythm overlap was substantial for each species pair (0.5 ≤ Δ < 1), the temporal segregation observed in their peak activity times may play a crucial role in partially mitigating interspecific competition and interference. These findings not only contribute to achieving a deeper understanding of interspecific relationships among sympatric species in the Qinghai–Xizang Plateau region but also contribute to establishing a scientific foundation for the conservation and management of large carnivores by emphasizing the significance of temporal niche partitioning.

## Figures and Tables

**Figure 1 animals-15-02059-f001:**
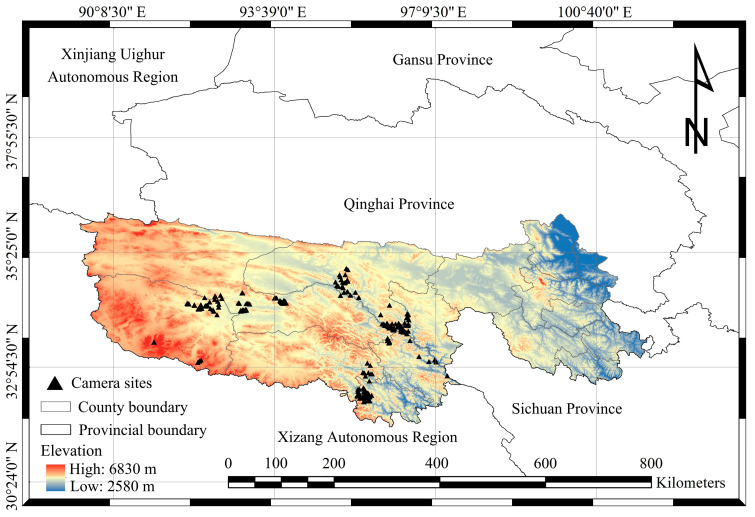
Locations of infrared camera trapping sites within the Sanjiangyuan Region.

**Figure 2 animals-15-02059-f002:**
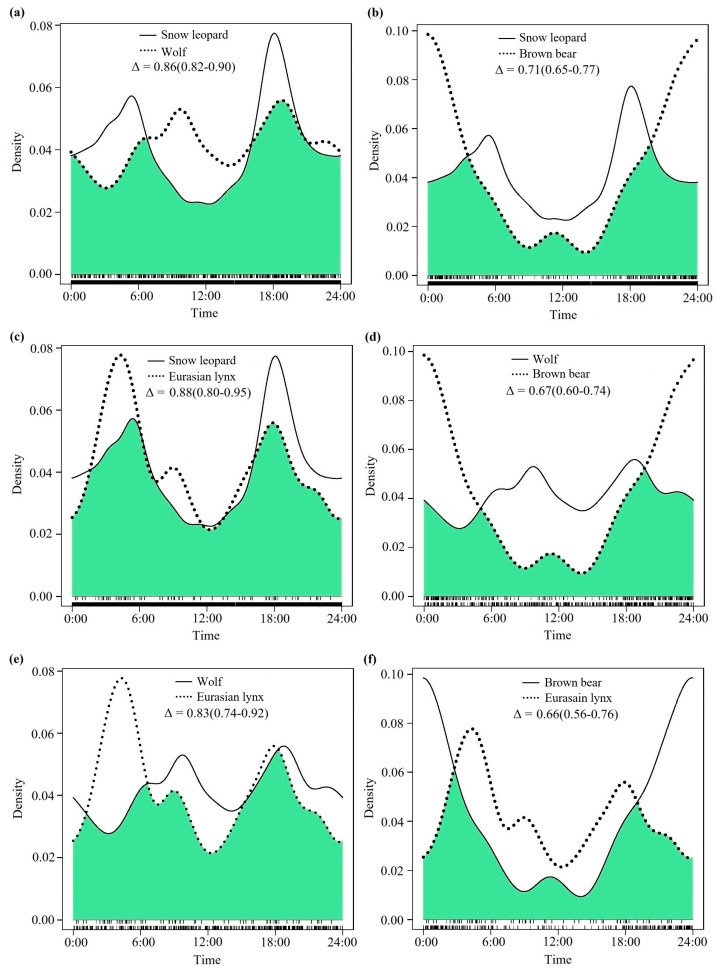
The comparative curves of diel activity rhythms between pairwise species among the four large carnivores throughout the year (January to December) in the SR. Δ represents the coefficient of overlap, while the numbers in parentheses represent the 95% confidence intervals for the coefficient of overlap (Δ). (**a**): Snow leopard and wolf; (**b**): snow leopard and brown bear; (**c**): snow leopard and Eurasian lynx; (**d**): wolf and brown bear; (**e**): wolf and Eurasian lynx; (**f**): brown bear and Eurasian lynx.

**Figure 3 animals-15-02059-f003:**
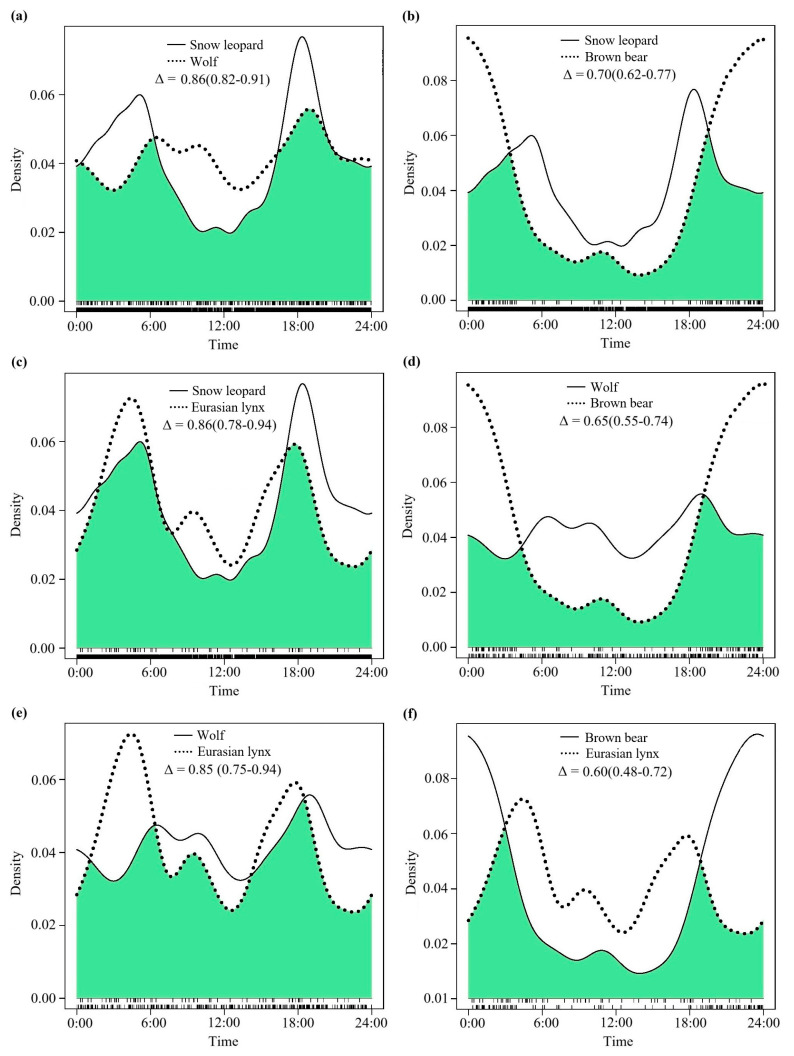
The comparative curves of diel activity rhythms in pairwise species among the four large carnivores during the cold season (October to April of the following year) in the SR. (**a**): Snow leopard and wolf; (**b**): snow leopard and brown bear; (**c**): snow leopard and Eurasian lynx; (**d**): wolf and brown bear; (**e**): wolf and Eurasian lynx; (**f**): brown bear and Eurasian lynx.

**Figure 4 animals-15-02059-f004:**
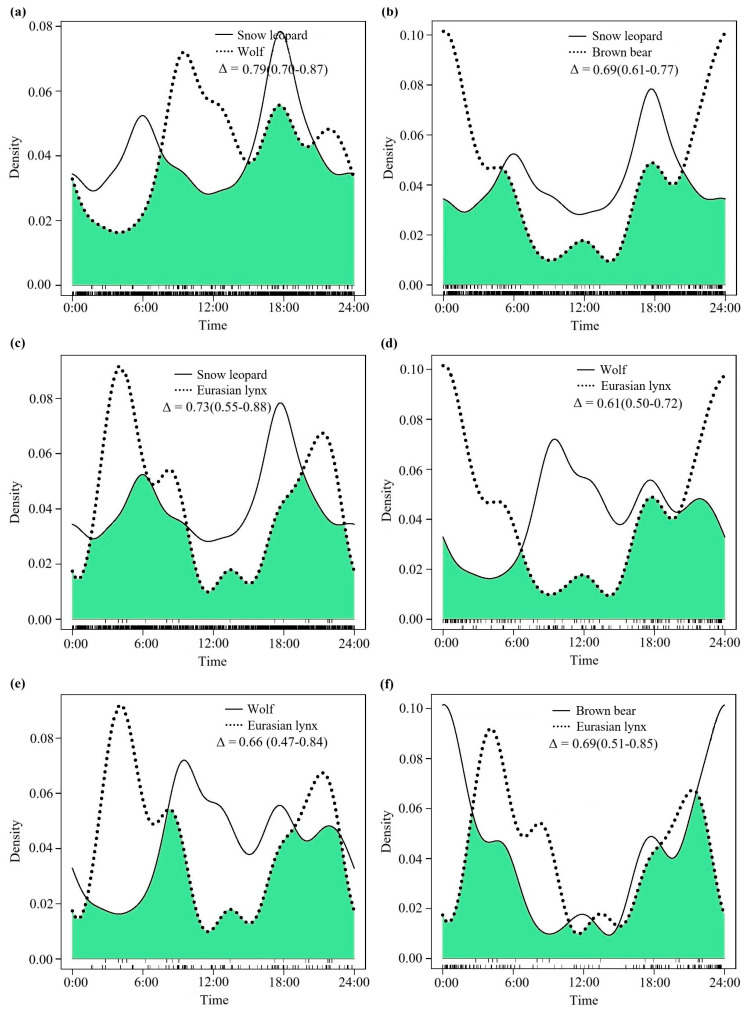
The comparative curves of diel activity rhythms in pairwise species among the four large carnivores during the warm season (May to September) in the SR. (**a**): Snow leopard and wolf; (**b**): snow leopard and brown bear; (**c**): snow leopard and Eurasian lynx; (**d**): wolf and brown bear; (**e**): wolf and Eurasian lynx; (**f**): brown bear and Eurasian lynx.

**Table 1 animals-15-02059-t001:** Table of survey information collected by infrared cameras across different regions and time periods in SR from June 2014 to April 2024.

Investigated Area	Survey Duration	No. of Camera Sites	Survey Area (km^2^)	Elevation Range (m)	Camera Days (d)
Yanzhanggua Valley	June 2014–December 2015	57	300	4125–5874	10,434
Jianggudiru Glacier	October 2015–October 2016	13	10	5432–6393	2143
Bagan Town	November 2018–September 2022	171	925	3897–4900	68,163
Longma Village	April 2022–April 2023	20	400	4102–5324	4781
Dongba Township	March 2023–March 2024	40	135	3982–4794	3106
Tanggulashan Town (Period 1)	September 2017–December 2018	60	60	4478–4979	4288
Tanggulashan Town (Period 2)	November 2023–April 2024	61	750	4507–5070	9998

**Table 2 animals-15-02059-t002:** Temporal overlap coefficients (Δ) of daily activity rhythms of large carnivore species pairs in Sanjiangyuan Region.

	*Panthera uncia-* *Canis lupus*	*Panthera uncia-* *Ursus arctos*	*Panthera uncia-* *Lynx lynx*	*Canis lupus-* *Ursus arctos*	*Canis lupus-* *Lynx lynx*	*Ursus arctos-* *Lynx lynx*
Annual	0.86 (0.82–0.90)	0.71 (0.65–0.77)	0.88 (0.80–0.95)	0.67 (0.60–0.74)	0.83 (0.74–0.92)	0.66 (0.56–0.76)
Cold season	0.86 (0.82–0.91)	0.70 (0.62–0.77)	0.86 (0.78–0.94)	0.65 (0.55–0.74)	0.85 (0.74–0.94)	0.60 (0.48–0.72)
Warm season	0.79 (0.70–0.87)	0.69 (0.61–0.77)	0.73 (0.55–0.88)	0.61 (0.50–0.72)	0.66 (0.47–0.84)	0.69 (0.51–0.85)

## Data Availability

The data are available upon reasonable request from the corresponding author (Xinming Lian).

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
