# Peer review of "Increased Temporal Overlap in Diel Activity Patterns Potentially Intensifies Interspecific Competition Among Sympatric Large Carnivores in the Sanjiangyuan Region of China"

_animals, 2025, doi:10.3390/ani15142059_

Round 1
Reviewer 1 Report
Comments and Suggestions for Authors
Comments
The manuscript explores the temporal overlap and interspecific competition among large carnivores in the Sanjiangyuan Region using long-term camera trap data. The findings contribute significantly to our understanding of how these apex predators coexist in a high-altitude ecosystem. While the study addresses an important topic in animal ecology, there are several areas where improvements can be made in terms of clarity, scientific rigor, and presentation. The following review comments are provided to assist in further refining the manuscript.
Abstract
- The abstract is generally well-structured, but it could benefit from more precise language. For instance. The abstract mentions "various monitoring intervals" but does not specify what these intervals are. This information is crucial for understanding the scope of the study.
Introduction
- I don’t think it’s necessary to elaborate on the role of camera traps. This section can be condensed to avoid redundancy. Highlighting the ecological issues related to your species is more important than discussing a well-established tool in ecological studies.
- The introduction provides a good overview of the importance of large carnivores in ecosystems. However, it lacks a clear definition of "temporal niche differentiation," which is central to the study. A brief definition and explanation of its ecological significance would improve clarity.
Materials and Methods
- The section on camera trapping and data pre-processing is thorough. However, the criteria for defining independent events (same species within a 30-minute interval) should be justified with references or explanations. This criterion is critical for data analysis, and its rationale should be clear.
- The use of kernel density estimation is appropriate, but the formula provided is not explained in detail. A brief explanation of the formula's components and their ecological significance would improve understanding.
Results
- The results section presents a large amount of data, but the presentation could be improved. For example, the overlap coefficients (Δ) are mentioned multiple times without a clear summary table or figure that consolidates these values for easy comparison.
- I would suggest adding animal icons when drawing the activity rhythm overlap graph. This can make the graph more intuitive and easier to understand.
Discussion
- The discussion mentions "temporal segregation likely functions to alleviate direct competition." This conclusion is speculative and should be supported with more evidence or references. The mechanisms by which temporal segregation reduces competition should be explained.
- The discussion refers to "body size significantly influences predation risk." This statement is not supported by data or references within the manuscript. It should either be backed up with specific examples or removed.
- The discussion concludes that coexistence is achieved through "temporal, spatial, and trophic niche differentiation." This conclusion is broad and should be supported with specific examples from the study or literature.
Conclusion
- The manuscript presents valuable data on the temporal activity patterns of large carnivores in the Sanjiangyuan Region. However, it requires revisions to improve clarity, scientific rigor, and presentation. The authors should address the issues highlighted above to ensure the manuscript meets the standards for publication.
References
- Please ensure that all citations are consistent with the chosen referencing style and that there are no missing references. Please check all the references and add the doi of each reference.

The quality of English language in this manuscript is generally good. The authors have effectively conveyed their research findings and ideas in a clear and coherent manner. The use of technical terms and jargon is appropriate for the subject matter, and the sentence structure is mostly grammatically correct. However, there are a few minor issues that could be improved. For example, some sentences could benefit from more concise wording to enhance readability. Additionally, there are occasional awkward phrasings or word choices that might be distracting to native English speakers. Overall, the language quality is satisfactory for publication, but a thorough proofreading by a native English speaker or a professional editor could further enhance the clarity and fluency of the manuscript.
Author Response
|
General comments: The manuscript explores the temporal overlap and interspecific competition among large carnivores in the Sanjiangyuan Region using long-term camera trap data. The findings contribute significantly to our understanding of how these apex predators coexist in a high-altitude ecosystem. While the study addresses an important topic in animal ecology, there are several areas where improvements can be made in terms of clarity, scientific rigor, and presentation. The following review comments are provided to assist in further refining the manuscript. |
|
Response: We sincerely appreciate your thorough and thoughtful feedback on our manuscript, which demonstrates your scientific rigor and has significantly contributed to the enhancement of our work. After carefully reviewing your suggestions, we have made substantial revisions, particularly in the methodology, results, discussion and conclusion sections, to address the raised issues. We believe that these modifications adequately address your concerns and provide a more comprehensive understanding of our study. Once again, we express gratitude for your valuable feedback and the opportunity it provided us to refine our manuscript. |
|
Detailed comments: |
|
Comments 1: Abstract. 1. The abstract is generally well-structured, but it could benefit from more precise language. For instance. The abstract mentions "various monitoring intervals" but does not specify what these intervals are. This information is crucial for understanding the scope of the study. |
|
Response 1: We appreciate your valuable feedback. Based on your suggestions, we have comprehensively revised the erroneous and ambiguous wording in the article's abstract to enhance its scientific rigor and readability. Please refer to lines 32-57 for further details. |
|
Comments 2-3: Introduction 2 I don’t think it’s necessary to elaborate on the role of camera traps. This section can be condensed to avoid redundancy. Highlighting the ecological issues related to your species is more important than discussing a well-established tool in ecological studies. 3. The introduction provides a good overview of the importance of large carnivores in ecosystems. However, it lacks a clear definition of "temporal niche differentiation," which is central to the study. A brief definition and explanation of its ecological significance would improve clarity. |
|
Response 2: We appreciate your insightful suggestion. This study primarily relies on infrared camera technology for monitoring four species of large carnivores. Therefore, we introduced the applications and significance of infrared cameras in the introduction section. Following your suggestion, we have condensed this part of the content. Please refer to lines 74-81 for further details. |
|
Response 3: Thank you so much for your meticulous work and invaluable comments. As you mentioned, temporal niche differentiation constitutes the core of this study, so we have added relevant descriptions on this aspect in the introduction section. Your revision suggestions have rendered the article more rigorous and substantial. Please refer to lines 82-105 for further details. |
|
Comments 4-5: Materials and Methods 4 The section on camera trapping and data pre-processing is thorough. However, the criteria for defining independent events (same species within a 30-minute interval) should be justified with references or explanations. This criterion is critical for data analysis, and its rationale should be clear. 5 The use of kernel density estimation is appropriate, but the formula provided is not explained in detail. A brief explanation of the formula's components and their ecological significance would improve understanding. |
|
Response 4: We appreciate your insightful suggestion. Indeed, the infrared camera photographs used to analyze the diurnal activity rhythms of the four large carnivores were all independent events. We have added references and provided supplementary information regarding the definition of independent events. Please refer to lines 190-195 for further details. |
|
Response 5: We appreciate your insightful suggestion. Based on your suggestions, we have elaborated on the ecological significance of kernel density estimation in describing animal diurnal activity rhythms and clarified the meanings of each parameter in its computational formula, aiming to enhance the readability and rigor of the article. Please refer to lines 197-202 and lines 206-211 for further details. |
|
Comments 6-7: Results 6. The results section presents a large amount of data, but the presentation could be improved. For example, the overlap coefficients (Δ) are mentioned multiple times without a clear summary table or figure that consolidates these values for easy comparison. 7. I would suggest adding animal icons when drawing the activity rhythm overlap graph. This can make the graph more intuitive and easier to understand. |
|
Response 6: Thank you so much for your meticulous work and invaluable comments. Following your suggestion, we have included the activity time overlap indices for species pairs across the entire year, cold season, and warm season in Table 2 within the Results section, facilitating easier reading and comparison. Please refer to lines 281-283 for further details. |
|
Response 7: Thank you for your invaluable comments. Following your suggestion, we attempted to incorporate corresponding animal icons into the activity rhythm overlap graphs. However, since each figure is composed of six merged subplots and the available space within each subplot is limited, the icons became illegible. Therefore, we have opted to represent the species using their Latin binomials instead. We appreciate your suggestion once again. |
|
Comments 8-10: Discussion 8. The discussion mentions "temporal segregation likely functions to alleviate direct competition." This conclusion is speculative and should be supported with more evidence or references. The mechanisms by which temporal segregation reduces competition should be explained. 9. The discussion concludes that coexistence is achieved through "temporal, spatial, and trophic niche differentiation." This conclusion is broad and should be supported with specific examples from the study or literature. |
|
Response 8: Thank you so much for your meticulous work. Based on your revision suggestions, we have added references to this section and supplemented the content accordingly. Please refer to lines 343-347 for further details. |
|
Response 9: We appreciate your insightful suggestion. As you pointed out, our study does not contain data supporting a significant influence of body size on predation risk or its role in interspecific competition. Therefore, following your suggestion, we have removed that paragraph to enhance the rigor of the paper. |
|
Response 10: Thank you so much for your meticulous work and invaluable comments. Following your suggestion, we have removed the paragraph discussing the role of body size in interspecific competition along with its broader content. The subsequent discussion now focuses primarily on the research findings. Additionally, in the final paragraph of the discussion, we have highlighted the limitations of this study, suggesting that future work should collect more data to explore the mechanisms of regional coexistence among the four large carnivores from the three ecological niche dimensions of space, time, and trophic level. Please refer to lines 326-381 for further details. |
|
Comments 11: Conclusion 11. The manuscript presents valuable data on the temporal activity patterns of large carnivores in the Sanjiangyuan Region. However, it requires revisions to improve clarity, scientific rigor, and presentation. The authors should address the issues highlighted above to ensure the manuscript meets the standards for publication. |
|
Response 11: Thank you for your invaluable comments. Following your suggestion, we have thoroughly revised and supplemented the discussion section and other parts of the manuscript to enhance its clarity, academic rigor, and readability. We sincerely appreciate your valuable feedback, which was crucial for improving the paper. Please refer to lines 388-399 for further details. |
|
Comments 12: References 12. Please ensure that all citations are consistent with the chosen referencing style and that there are no missing references. Please check all the references and add the doi of each reference. |
|
Response 12: Thank you so much for your meticulous work and invaluable comments. We meticulously reviewed each reference entry, ensuring the accuracy and standardization of citations before uniformly formatting all the references. Additionally, following your suggestion, we have appended a unique DOI number to each reference. Please refer to lines 414-554 for further details. |
|
Response to Comments on the Quality of English Language |
|
Point 1: The quality of English language in this manuscript is generally good. The authors have effectively conveyed their research findings and ideas in a clear and coherent manner. The use of technical terms and jargon is appropriate for the subject matter, and the sentence structure is mostly grammatically correct. However, there are a few minor issues that could be improved. For example, some sentences could benefit from more concise wording to enhance readability. Additionally, there are occasional awkward phrasings or word choices that might be distracting to native English speakers. Overall, the language quality is satisfactory for publication, but a thorough proofreading by a native English speaker or a professional editor could further enhance the clarity and fluency of the manuscript. |
|
Response 1: Thank you for your feedback regarding the quality of English in our manuscript. We appreciate your understanding and will strive to improve the clarity and precision of our language in the revised manuscript. All modifications throughout the manuscript are highlighted in red font. |

Reviewer 2 Report
Comments and Suggestions for Authors
Dear author(s),
The manuscript is composed effectively and serves as a valuable resource for the international audience and scientific community. Below, I have attempted to highlight several detailed review reports regarding the manuscript:
- The author attempted to tackle the heightened temporal overlap in diel activity patterns, which intensifies interspecific competition among sympatric large carnivores in the Sanjiangyuan Region of China. Nevertheless, it would also be beneficial to illustrate the dietary preferences of these four carnivores.
- I am of the opinion that the subject matter is both original and pertinent to the discipline. Furthermore, it is anticipated to address a gap within the field.
- In contrast to previous studies conducted on the four sympatric large carnivores in the SR, which have primarily concentrated on dietary composition, predator-prey interspecific relationships, human-carnivore conflict, and population density, the current study aims to deepen the understanding of interspecific competitive relationships among these sympatric large carnivores. This research utilizes a decade-long dataset derived from infrared camera trap records.
- In my view, the study employed all necessary methodological procedures and analytical tools to uncover significant findings for this research project. However, the aforementioned limitation pertains to illustrating the dietary preferences of the four carnivores.
- The conclusion presented aligns with the findings and addresses the primary research question. The research yielded a valid conclusion, and the work is underpinned by adequate data.
- Nearly all of the reference lists are suitable and current. Nevertheless, there are a few outdated sources that should be substituted with more recent studies.
- All tables and figures are presented suitably and in an appropriate manner.
- I provided several editorial and clarity suggestions regarding the reviewed document.
Best regards,

I provided several editorial and clarity suggestions regarding the reviewed document. Please make improvements to the document based on these recommendations.
Author Response
|
General comments: The manuscript is composed effectively and serves as a valuable resource for the international audience and scientific community. Below, I have attempted to highlight several detailed review reports regarding the manuscript: 1. The author attempted to tackle the heightened temporal overlap in diel activity patterns, which intensifies interspecific competition among sympatric large carnivores in the Sanjiangyuan Region of China. Nevertheless, it would also be beneficial to illustrate the dietary preferences of these four carnivores. 2. I am of the opinion that the subject matter is both original and pertinent to the discipline. Furthermore, it is anticipated to address a gap within the field. 3. In contrast to previous studies conducted on the four sympatric large carnivores in the SR, which have primarily concentrated on dietary composition, predator-prey interspecific relationships, human-carnivore conflict, and population density, the current study aims to deepen the understanding of interspecific competitive relationships among these sympatric large carnivores. This research utilizes a decade-long dataset derived from infrared camera trap records. 4. In my view, the study employed all necessary methodological procedures and analytical tools to uncover significant findings for this research project. However, the aforementioned limitation pertains to illustrating the dietary preferences of the four carnivores. 5. The conclusion presented aligns with the findings and addresses the primary research question. The research yielded a valid conclusion, and the work is underpinned by adequate data. 6. Nearly all of the reference lists are suitable and current. Nevertheless, there are a few outdated sources that should be substituted with more recent studies. 7. All tables and figures are presented suitably and in an appropriate manner. 8. I provided several editorial and clarity suggestions regarding the reviewed document. Best regards. |
|
Response: Thank you for your affirmation of our article and for recognizing the improvements made to the manuscript. We have carefully considered your suggestions and made detailed changes. Once again, thank you for your valuable feedback and the opportunity to refine our manuscript. |
|
Detailed comments: |
|
Comments 1: L3-L4: It was also better to show the diet preference of those four carnivores. |
|
Response 1: We appreciate your valuable suggestion. However, this study primarily explores interspecific interference and competition based on the degree of inter-specific overlap in the diurnal activity rhythms of four large carnivores in the SR, without involving dietary preferences. Therefore, we retain the original title. We sincerely appreciate your suggestions once again. |
|
Comments 2: L19-L20: Complex predator guilds further intensify these competitive pressures with dominance hierarchies that clearly define winners and losers. Replace with the above statement. |
|
Response 2: We express our gratitude to your valuable comment. Following your suggestions, we have rephrased this sentence. Please see clear version of manuscript: lines 20-21 for details. |
|
Comments 3: L27: combination is modified to combinations. |
|
Response 3: Thank you so much for your careful work. We apologize for those mistakes. We have made the correction from combination to combinations. Please see clear version of manuscript: line 29 for details. |
|
Comments 4: L37: 3312 is modified to 3,312. |
|
Response 4: Thank you so much for your careful work. We have reviewed the formatting guidelines of the journal Animals, and noted that numbers exceeding 1,000 are not separated by commas. |
|
Comments 5: L9: add comma before respectively. |
|
Response 5: Thank you so much for your careful work. We added a comma before respectively. |
|
Comments 6: L42-43: overlap is modified to overlaps; intensity is modified to intensities; four is modified to those four. |
|
Response 6: Thank you for your meticulous work and valuable feedback. According to your revision suggestions, we have made all the necessary modifications to the relevant content. Please refer to lines 43-45 for further details. |
|
Comments 7: L43: delete the list of animals as they are already listed above. |
|
Response 7: We appreciate your valuable suggestion. Following your suggestions, we have removed the list of the names of these four animals. |
|
Comments 8: delete comma. |
|
Response 8: Thank you for your meticulous work and invaluable comments. We removed the comma. |
|
Comments 9: L56-L57: The yellow dragged words/phrases are repetition of the topics. Pls., replace by other equivalent words/phrases. |
|
Response 9: Thank you for your meticulous work and invaluable comments. Following your suggestion, we have rewritten the keywords to avoid repetition with the title. Please refer to lines 58-59 for further details. |
|
Comments 10: L67-L68: public perceptions and attitudes. |
|
Response 10: We appreciate your valuable suggestion. Following your suggestions, we have revised "public perceptions, attitudes" to "public perceptions and attitudes". Please refer to line 70 for further details. |
|
Comments 11: L72: delete comma. |
|
Response 11: Thank you for your meticulous work and invaluable comments. We removed the comma. |
|
Comments 12: L104: Remove it from here and put it at the end of the statement. |
|
Response 12: We appreciate your valuable suggestion. Following your suggestions, we have relocated this part of the content to the end of the sentence. Please refer to lines 133-134 for further details. |
|
Comments 13: L105-L106: covering is modified to covers; revers is modified to rivers. |
|
Response 13: Thank you for your meticulous work and invaluable comments. We apologize for those mistakes. We have corrected the incorrectly expressed words. Please refer to lines 133-134 for further details. |
|
Comments 14: L121-L123: needs sources. |
|
Response 14: Thank you so much for your careful work. We have added references for the corresponding content. |
|
Comments 15: L126-L127: Why you selected only these four carnivores for this study? It is better to show the selection criteria. How many of the 85 mammalian species of the region are large carnivores? |
|
Response 15: We appreciate your valuable suggestion. Following your suggestions, we reviewed relevant literature, supplemented the list of large carnivore species in the SR, and explained the rationale for selecting snow leopards, wolves, brown bears, and lynxes as the focal study species among all large carnivores in the region. Please refer to lines 156-158 for further details. |
|
Comments 16: L127-L128: What was the distance between those cameras? 500m? |
|
Response 16: Thank you for your meticulous work and invaluable comments. As you mentioned, the deployment spacing between adjacent infrared cameras is greater than 500 meters, and we have incorporated this point into our infrared camera deployment plan. Please refer to lines 177-180 for further details. |
|
Comments 17: L139, L143, L145, L146, and L151: oriented is modified to orientated; information followed by a comma; characteristics followed by a comma; excessively is modified to excessive; site is modified to sites. |
|
Response 17: Thank you for your meticulous work and valuable feedback. We apologize for those mistakes. We have made revisions to address the specific details and spelling errors you pointed out. We sincerely appreciate your meticulous and thorough feedback. |
|
Comments 18: L196: about ten years? |
|
Response 18: We appreciate your valuable suggestion. We apologize for the error that has arisen. The correct time frame should be from June 2014 to April 2024. Please refer to line 205 for further details. |
|
Comments 19: L213, L220, and L242: delete moderate; represented is modified to represent; remove comma after season. |
|
Response 19: Thank you so much for your careful work. We have made revisions to address the specific details and spelling errors you pointed out. Your revisions to the details have significantly enhanced the quality of our article. |
|
Comments 20: Discussion, L248, L267, L278, L315, L329: mountains is modified to mountain; influence is modified to influenced; divers is modified to diver; inhabit is modified to inhabits; evolutionary is modified to the evolutionary. |
|
Response 20: Thank you for your meticulous work and valuable feedback. We apologize for those mistakes. We have made revisions to all the aforementioned detailed issues you raised. |
|
Comments 21: Conclusions, L343: declined is modified to declines; are is modified to is. |
|
Response 21: Thank you so much for your careful work. We have made corrections to both of the word modification suggestions you proposed. Your revisions to the details have significantly enhanced the quality of our article. |
|
Response to Comments on the Quality of English Language |
|
Point 1: The English could be improved to more clearly express the research. |
|
Response 1: Thank you for your feedback regarding the quality of English in our manuscript. We appreciate your understanding and will strive to improve the clarity and precision of our language in the revised manuscript. All modifications throughout the manuscript are highlighted in red font. |

Reviewer 3 Report
Comments and Suggestions for Authors
Overall Evaluation:
Monitoring large carnivores presents considerable challenges due to the inaccessibility of their natural habitats and their elusive behavior. Over a period spanning nearly a decade, this study employed infrared camera trapping technology to collect an extensive dataset of images capturing snow leopards, wolves, brown bears, and Eurasian lynx, an achievement that constitutes a meaningful contribution to the field. The authors investigated the intensity of interspecific competition and species interactions from the perspective of temporal niche partitioning, offering compelling empirical support. Additionally, the paper is logically structured, includes a thorough and methodical discussion, and its findings serve as a valuable foundation for future research into the mechanisms that facilitate coexistence among large carnivores in high-altitude ecosystems.
Detailed comments:
In the original text, each species should be accompanied by its full binomial nomenclature upon first mention, including the author and year of description, rather than being referred to solely by its Latin name. For example: (Vulpes vulpes Linnaeus, 1758). This ensures taxonomic clarity and facilitates accurate scientific communication.
L79, L93: Add reference.
L103: A concise overview of the approximate geographic distribution of the four large carnivore species within the Sanjiangyuan Region should be included in the description of the study area.
L116-117: In the context of this study, it is essential to explicitly define the specific months corresponding to the cold and warm seasons within the Sanjiangyuan Region by providing a clear temporal range.
L200-201: The authors are encouraged to explicitly report the number of effective detection events (or valid records) for each of the following species in the manuscript: snow leopard (Panthera uncia), wolf (Canis lupus), brown bear (Ursus arctos), and Eurasian lynx (Lynx lynx).
L209: The acronym CI should be explicitly defined upon its first use in the manuscript.
L303-306: While this study has focused exclusively on the role of temporal niche differentiation in the regional coexistence of the four large carnivore species, spatial and trophic niches are also mentioned. Therefore, it is recommended that the authors review relevant literature and, in combination with the findings of this study, undertake a comprehensive analysis of interspecific relationships and regional coexistence mechanisms across the three niche dimensions: temporal, spatial, and trophic level.
In conclusion, I believe that the manuscript will meet publishable standards after the aforementioned revisions are implemented.
Author Response
|
General comments: Monitoring large carnivores presents considerable challenges due to the inaccessibility of their natural habitats and their elusive behavior. Over a period spanning nearly a decade, this study employed infrared camera trapping technology to collect an extensive dataset of images capturing snow leopards, wolves, brown bears, and Eurasian lynx, an achievement that constitutes a meaningful contribution to the field. The authors investigated the intensity of interspecific competition and species interactions from the perspective of temporal niche partitioning, offering compelling empirical support. Additionally, the paper is logically structured, includes a thorough and methodical discussion, and its findings serve as a valuable foundation for future research into the mechanisms that facilitate coexistence among large carnivores in high-altitude ecosystems. |
|
Response: We sincerely appreciate your thorough and thoughtful feedback on our manuscript, which demonstrates your scientific rigor and has significantly contributed to the enhancement of our work. Once again, we express gratitude for your valuable feedback and the opportunity it provided us to refine our manuscript. The explanations and modifications we have provided in response to the several questions/doubts you raised are as follows. |
|
Detailed comments: |
|
Comments 1: In the original text, each species should be accompanied by its full binomial nomenclature upon first mention, including the author and year of description, rather than being referred to solely by its Latin name. For example: (Vulpes vulpes Linnaeus, 1758). This ensures taxonomic clarity and facilitates accurate scientific communication. |
|
Response 1: Thank you so much for your meticulous work. We have revised all the first occurrences of species’ Latin scientific names throughout the text, representing them with the Latin name, the namer, and the year of naming. |
|
Comments 2: L79, L93: Add reference. |
|
Response 2: Thank you so much for your meticulous work. Following your suggestion, we have added references to these two sentences respectively. |
|
Comments 3: L103: A concise overview of the approximate geographic distribution of the four large carnivore species within the Sanjiangyuan Region should be included in the description of the study area. |
|
Response 3: Thank you for your valuable feedback. We included the snow leopard, wolf, brown bear and lynx in the infrared camera trap deployment The main reason for choosing snow leopards, wolves, brown bears and lynx as the research subjects in this study is that the four large carnivores are more common and sympatrically distributed in the Sanjiangyuan Region. Please refer to lines 156-158 for more details. |
|
Comments 4: L116-117: In the context of this study, it is essential to explicitly define the specific months corresponding to the cold and warm seasons within the Sanjiangyuan Region by providing a clear temporal range. |
|
Response 4: We appreciate your insightful suggestion. In the article, we reviewed relevant literature and provided clear definitions for the time periods of the cold season and warm season in the Sanjiangyuan Region. Please refer to lines 212-214 for more details. |
|
Comments 5: L200-201: The authors are encouraged to explicitly report the number of effective detection events (or valid records) for each of the following species in the manuscript: snow leopard (Panthera uncia), wolf (Canis lupus), brown bear (Ursus arctos), and Eurasian lynx (Lynx lynx). |
|
Response 5: Thank you so much for your meticulous work and invaluable comments. Following your suggestion, we have incorporated valid records of the four large carnivores obtained throughout the entire survey period into our research findings. Please refer to lines 237-239 for more details. |
|
Comments 6: L209: The acronym CI should be explicitly defined upon its first use in the manuscript. |
|
Response 6: Thank you for your valuable feedback. The specific meaning represented by the first occurrence of "CI" has been added to our manuscript. |
|
Comments 7: L303-306: While this study has focused exclusively on the role of temporal niche differentiation in the regional coexistence of the four large carnivore species, spatial and trophic niches are also mentioned. Therefore, it is recommended that the authors review relevant literature and, in combination with the findings of this study, undertake a comprehensive analysis of interspecific relationships and regional coexistence mechanisms across the three niche dimensions: temporal, spatial, and trophic level. |
|
Response 7: Thank you so much for your meticulous work and invaluable comments. Following your suggestion, upon reviewing the literature, we found that no single niche differentiation mechanism can fully explain the regional coexistence of large carnivores. In the discussion section of the article, we also incorporated a discussion on the spatial utilization and dietary differences among the four large carnivores. Furthermore, we acknowledged the limitations of this study, suggesting that future research should explore the mechanisms of regional coexistence among large carnivores from multiple niche dimensions. Please refer to lines 326-381 for more details. |
|
Response to Comments on the Quality of English Language |
|
Point 1: The English is fine and does not require any improvement. |
|
Response 1: Thank you for your recognition of our entire manuscript writing. We have conducted a thorough review of the full text and made revisions to the unclear and erroneous parts in order to enhance the rigor and readability of the article. All modifications throughout the manuscript are highlighted in red font. |

Round 2
Reviewer 1 Report
Comments and Suggestions for Authors Thank you for your recent submission. I appreciate the effort you have put into your work. However, I would like to bring a couple of issues to your attention that need to be addressed. Firstly, the preface of your manuscript is rather lengthy. While it provides a comprehensive background, some parts may overlap or lack tight cohesion. I suggest revising it by merging closely related sections to streamline the content. This will help maintain the reader’s focus and make the introduction more impactful. Secondly, there seems to be a misunderstanding regarding Question 7. The issue is not related to the table but rather to the legend of Figure 4. The suggestion is to consider adding the appropriate species names in the legend of Figure 4 to enhance the visual appeal and clarity of the illustration. This will make it easier for readers to understand the context of the figure without needing to refer to the table. I hope these comments are helpful. I look forward to seeing the revised version of your manuscript. Comments on the Quality of English LanguageFirstly, the preface of your manuscript is rather lengthy. While it provides a comprehensive background, some parts may overlap or lack tight cohesion. I suggest revising it by merging closely related sections to streamline the content. This will help maintain the reader’s focus and make the introduction more impactful.
Author Response
|
General comments: Thank you for your recent submission. I appreciate the effort you have put into your work. However, I would like to bring a couple of issues to your attention that need to be addressed. Firstly, the preface of your manuscript is rather lengthy. While it provides a comprehensive background, some parts may overlap or lack tight cohesion. I suggest revising it by merging closely related sections to streamline the content. This will help maintain the reader’s focus and make the introduction more impactful. Secondly, there seems to be a misunderstanding regarding Question 7. The issue is not related to the table but rather to the legend of Figure 4. The suggestion is to consider adding the appropriate species names in the legend of Figure 4 to enhance the visual appeal and clarity of the illustration. This will make it easier for readers to understand the context of the figure without needing to refer to the table. I hope these comments are helpful. I look forward to seeing the revised version of your manuscript. |
|
Response: We sincerely appreciate your positive evaluation of our article and your acknowledgment of the improvements made to the manuscript. We have carefully reviewed your suggestions and implemented detailed revisions accordingly. Once again, we extend our gratitude for your valuable feedback and the opportunity to further enhance the quality of our work. |
|
Response 1: Thank you for your thoughtful suggestion. In response to your recommendation, we have thoroughly revised and improved the introductory section of the manuscript. Your input has significantly strengthened the clarity and academic rigor of our work. For reference, please see lines 62–109. Response 2: We are grateful for your meticulous review and constructive comments. We sincerely apologize for the errors identified. As per your recommendations, we have added species names to the legends of Figures 2, 3, and 4 to improve the readability and contextual clarity of the figures. Please refer to lines 242–244, 259–261, and 264–267 for the corresponding modifications. |
|
Response to Comments on the Quality of English Language |
|
Point 1: Firstly, the preface of your manuscript is rather lengthy. While it provides a comprehensive background, some parts may overlap or lack tight cohesion. I suggest revising it by merging closely related sections to streamline the content. This will help maintain the reader’s focus and make the introduction more impactful. |
|
Response 1: Thank you for your valuable feedback concerning the language quality of our manuscript. In accordance with your suggestions, we have thoroughly revised and improved the introductory section of the paper. |
